# PERFORMANCE ADJUSTMENT FOR FEDERATED LEARNING MARKETPLACE

## ABSTRACT

In federated learning, client participation is mainly motivated by *performance-gain rewards* or *monetary rewards*. In practice, different clients may have varying preferences over these two types of rewards. However, optimizing the training process to align model performance and monetary rewards with client expectations remains an open challenge. To accommodate diverse reward preferences, we propose Alpha-Tuning, an FL performance adjustment framework guided by dynamic validation loss composition. The core of our framework is a mechanism to decide the weights assigned to clients' local validation loss, each of which is determined by the corresponding client's performance contribution in the given training round and its monetary quotation for biasing this FL course towards its favor. The training hyper-parameters and model aggregation weights are adjusted together with model parameters to minimize the weighted sums of clients' local validation losses in our framework. Paired with a payment rule designed to compensate the clients according to their data contribution, Alpha-Tuning balances the clients' preferences between the performance gain and monetary reward. We demonstrate the effectiveness of our framework by conducting experiments on the federated learning tasks under various client quotation settings.

## 1 INTRODUCTION

Recently, federated learning (FL) has garnered significant attention due to its capability to collaboratively train models from various isolated data sources without the direct sharing of private data (Kairouz et al., 2021). The success of an FL greatly depends on the active participation of clients that possess valuable data. As clients engage in FL primarily to achieve performance gains, monetary rewards, or both (Pei, 2020; Zeng et al., 2021), the appropriate incentive mechanism becomes an essential component of FL.

Most existing incentive mechanisms take a post-hoc form, where clients receive monetary rewards based on their data contribution after the trained model has generated revenue (Wang et al., 2020). However, such incentives may not be flexible in cases where clients act as both data contributors and model buyers with different preferences towards rewards. For example, if a client with less data is willing to pay more to obtain a better model performance gain, in the post-hoc form, there is no way to transfer the monetary investment to the model performance gain in the FL training process, since in such cases, clients can only influence model training through their data. Moreover, some clients act as data merchants, and their primary objective is to obtain monetary rewards by contributing their data to FL training, regardless of the model performance they obtain. Therefore, the challenge is to design an FL framework with an embedded incentive mechanism that can adaptively adjust different clients' preferences towards both monetary rewards and model performance gain rewards.

In light of the federated hyper-parameter optimization (FedHPO) Khodak et al. (2021); Guo et al. (2022); Wang et al. (2023), which efficiently searches for the suitable hyper-parameter configurations, we propose a novel FL framework, named as Alpha-Tuning. The core of our proposed framework is the MarketFedHPO component, which enables the adjustment of the performance of the model during the FL training process according to clients' willingness. The willingness, named as alpha value, is measured by the overall contribution that is a combination of clients' monetary quotation (monetary contribution) and their data value (data contribution). With the overall-contribution based willingness, the performance adjustment is achieved by prioritizing the model performance of clients

who have made higher willingness through the hyper-parameter selection criteria. By using this overall contribution-based hyper-parameter selection criteria, clients with lower-quality data can pursue a higher alpha value by submitting a higher bid/quotation, thereby obtaining a model that focuses on improving performance on their data. After training, the payment schema allocates the entire amount in the bidding pool among clients based on their data contribution. Those clients who have provided high-quality data are rewarded with a greater amount of monetary compensation through this payment schema. The combination of the overall contribution-based FedHPO and data contribution-based payment schema allows for a flexible approach to satisfying clients' varying preferences regarding model performance gain and monetary reward. Our experiments with different client quotation settings demonstrate the effectiveness of this proposed method in terms of preference adjustment.

## 2 BACKGROUND AND RELATED WORK

**Federated Learning (FL).** Federated learning (Kairouz et al., 2021) is a framework designed for facilitating the collaboration on training machine learning models within a multi-client setting and without directly sharing clients' local datasets. Formally speaking, assume there is a set of clients $\mathcal{C} = \{1, \ldots, N\}$ in the FL training. Each client $i \in \mathcal{C}$ has its own dataset $D^i \sim (\mathcal{X}^i)^{n_i}$, where $\mathcal{X}^i$ is the local data distribution and $n_i$ is the number of local samples. An important concern in FL is data heterogeneity, which means that $\mathcal{X}^i$ and $\mathcal{X}^j$ are probably two different distributions if $i \neq j$. We further denote the virtual global dataset as $D^{gl} = \{D^1, \ldots, D^N\}$. Similar to the standard procedure in traditional machine learning, the client divides its local dataset into training, validation and testing sets, denoted as $D_T^i, D_V^i$ and $D_E^i$. In this paper, we only focus on the properties of the global aggregated model $\mathbf{w}$ in FL. By taking a model $\mathbf{w}$ and a dataset $D$ as input, the loss function is denoted as $L(\mathbf{w}, D)$. The classic FL training can be formalized as the following empirical risk minimization problem:

$$\min_{\mathbf{w} \in \mathcal{W}} \sum_{i \in \mathcal{C}} \beta_i L(\mathbf{w}, D_T^i). \tag{1}$$

We call the $\beta_i$ as *training aggregation weights*. The most classic setting is $\beta_i = \frac{n_i}{n}$, which is equivalent to minimize the loss according to the virtual global dataset $D^{gl}$. However, the training aggregation weights are subject to changes for different tasks.

**Federated hyper-parameter optimization.** Khodak et al. (2021) introduce and compare federated successive halving algorithm (SHA) and FedEx. A similar method (Guo et al., 2022) is also proposed to accomplish hyper-parameter optimization in one trial, but update the hyper-parameter selection model with additional rounds of communication. A recent paper (Zhou et al., 2023) was proposed to search for good hyper-parameters before the FL training. Besides, there are other FedHPO with a different focus or different setting, including by extrapolation on learning rate (Koskela & Honkela, 2018), by Bayesian optimization (Dai et al., 2020), by representation matching (Mostafa, 2019) or by optimizing from system perspective (Zhang et al., 2023).

In recent FedHPO studies, a common routine is to take the hyper-parameters $\theta$ (i.e., local learning rate and local training iterations) of optimization algorithms (i.e., stochastic gradient descent) as additional trainable parameters. If we denote the loss function with hyper-parameters as $L((\mathbf{w}, \theta), D)$, the federated hyper-parameter optimization problem can be formalized as:

$$\min_{\theta \in \Theta} \sum_{i \in \mathcal{C}} \alpha_i L_{\text{hyper}}\left((\mathbf{w}, \theta), D_V^i\right) \text{ s.t. } \mathbf{w} \in \arg\min_{\mathbf{u}} \sum_{i \in \mathcal{C}} \beta_i L_{\text{train}}\left((\mathbf{u}, \theta), D_T^i\right). \tag{2}$$

The above problem formulation contains the training of model parameters on training sets as the condition, and places the hyper-parameter choosing step with validation set in the objective. Solving the problem can output the best hyper-parameter based on the best model performance each hyper-parameter can produce. Notice that $L_{\text{train}}$ and $L_{\text{hyper}}$ are not necessarily the same. While $L_{\text{train}}$ are usually traditional loss function (e.g., cross-entropy loss) for the need of applying efficient optimization algorithm (e.g., stochastic gradient descent), the $L_{\text{hyper}}$ can be more flexible and customize for different purposes. For example, when both precision and recall are important, $L_{\text{hyper}}$ can be the negated F1 score. Also, notice that the training aggregation weights $\beta_i$ and the weights in composing validation results $\alpha_i$ can be different. Especially, we can treat the $\beta_i$ as part of the tunable hyper-parameters while $\alpha_i$ are designed to favor specific clients.

If we embrace the spirit of Bayesian, the algorithm can be designed as the hyper-parameters $\theta$ are sample from a distribution $P_\gamma$ parametered by $\gamma$. It can be further reduced to a "single-level" empirical risk minimization problem (Li et al.; Khodak et al., 2021)

$$\min_{\gamma \in \Gamma, \mathbf{w} \in \mathcal{W}} \mathbb{E}_{\theta \sim P_\gamma} \left[ \sum_{i \in \mathcal{C}} \alpha_i L_{\mathsf{hyper}} \left( (\mathbf{w}, \theta), D_V^i \right) + \beta_i L_{\mathsf{train}} \left( (\mathbf{w}, \theta), D_T^i \right) \right]. \qquad (3)$$

After such transformation, the problem can be solve by updating the $\gamma$ and $\mathbf{w}$ alternatively.

**Data market and valuation of federated learning.** To provide a more direct incentive for the client to participate in the FL training, many researchers have explored pricing strategies for data sharing and federated learning (Pei, 2020; Zeng et al., 2021). Although numerical technical routines have demonstrated their potential, the most prevalent approaches are based on Shapley value (Ghorbani & Zou, 2019; Song et al., 2019; Wang et al., 2019; Liu et al., 2021) and auction mechanisms (Agarwal et al., 2019; Deng et al., 2021; Zeng et al., 2020). Our solution shares partial similarities with both approaches: our influence score is similar to Shapley value, but simplified to save the computation and communication cost in FL; the client quotation inherits the spirit of auction in the sense that the higher the price a client pays, the more the model favors it. But most of the existing work derives the monetary results afterward, while the monetary quotations in our solution affect the training on the fly together with contribution scores.

## 3 PERFORMANCE ADJUSTMENT VIA WEIGHTED VALIDATION LOSSES

**Problem formulation.** In practice, due to the heterogeneous data distribution among participants in FL, models may exhibit varying performance on clients' local data. Participants in FL may have incentives to influence the model training process to exert more substantial influence over the final model and increase model performance on their own data distribution. On the contrary, some participants may care less about the final model performance on their local datasets but want to profit from their contribution to the global model. Thus, a problem arises as how to design a dynamic mechanism to satisfy the FL participants in terms of both *local model performance* (e.g., test accuracy on local testing set) and *potential monetary payments or rewards*.

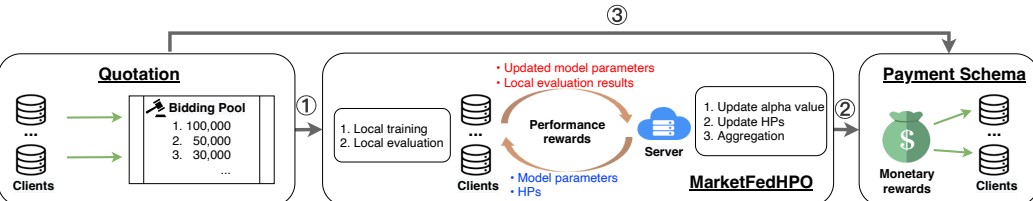

Figure 1: Framework of Alpha-Tuning. HPs stands for hyper-parameters.

**Overview of our solution: Alpha-Tuning.** The primary contribution of this paper lies in exploring an effective and fair approach to incentivize the FL training. As shown in Fig. 1, like many traditional FL algorithms, the local model parameters are updated with classic optimization algorithms, such as stochastic gradient descent (SGD) to ensure the model is optimized towards better solutions on the local training set. To build a bridge between the model performance preferences and the monetary quotation, we introduce a mechanism to quantify the influence of clients on the model training process. The mechanism updates a set of weights denoted by $\boldsymbol{\alpha}$, and the weighted sums of the local validation losses with $\boldsymbol{\alpha}$ guide the selection of 1) the aggregation weights when server merges the local updates with FedAvg, and 2) the hyper-parameters in the local training. Finally, a payment method re-distributed the total monetary quotations of all clients as payments or rewards. Eventually, our framework achieves "what you pay is what you get" regarding the final model performance and monetary payments and rewards. We will introduce the details of our framework, including how to quantify the performance contribution and monetary quotation, how to compute $\boldsymbol{\alpha}$ with these two kinds of variables, and how to use the $\boldsymbol{\alpha}$ to balance the performance gain rewards and monetary rewards.

### 3.1 PERFORMANCE GAIN METRIC: MUTUAL BENEFICIAL SCORE

When designing the metric for performance gain in the federated learning setting, we want to ensure the metric has the following properties.

*1) Compatible with different local optimization and global aggregation algorithms.* Since we focus on balancing the performance gain and monetary reward among clients, the optimal local training and global aggregation strategy should be left for clients to decide. However, notice that the model performance gain may differ for clients with different local training and global aggregation strategies. A desideratum is that such a metric can reflect performance contribution with different algorithms, but the metric score is also consistent with the algorithm chosen.

*2) Efficient in terms of communication cost and computation cost for training and evaluation.* In the FL setting, the communication cost of transferring models between servers and clients is a major factor in efficiency concerns. Meanwhile, as the client's computation power may be limited, we also need to consider the computation cost of evaluation. If the framework introduces a large number of additional communication rounds and local evaluations, it will be unwelcome in practice.

*3) Clear in mutual beneficial relations and appreciative for the improvement because of heterogeneous data, not similar data.* This property echoes FL's original goal: to take advantage of heterogeneous data to improve a model. Assume that in the data market setting, Client A, as a buyer, is interested in paying for co-train models with Client B. A reasonable outcome is that if Client B's local data is the same as Client A's, Client A should be obligated to pay. Besides, because we want to enable the monetary flow between clients, clear mutual beneficial relations are required for reasoning.

*4) Flexible enough for different axes.* The clients' performance focus may shift as the machine learning task changes. In Click-through-rate prediction tasks, the data may be very imbalanced, and improving the true positive may be the most desirable. However, minimizing the false negative may be the most crucial goal if a task is detecting cancer from MRI images.

To satisfy the above properties when quantifying the performance contribution, we introduce *mutual beneficial score (MBS)*, denoted as $B_{i \to k}^{(t)}$. Intuitively, it follows the spirit of leave-one-out (LOO) (Cook, 1977), measuring the impact of client $i$ to the client $k$'s local evaluation. The higher $B_{i \to k}^{(t)}$ is, client $i$ makes a larger contribution to improve the performance on client $k$'s local dataset.

**Computation of MBS.** Following the spirit of LOO, the FL training will actually train $N + 1$ model. One model is trained with all clients; each of the other $N$ models is trained with updates from $N - 1$ clients. We denote $\mathbf{w}^{(t)}$ as the model at the $t$-th FL training round with contributions from all clients. Symmetrically, we use $\mathbf{w}_{\mathcal{C}/i}^{(t)}$ to denote the model at the $t$-th FL round, and without involving client $i$ from the beginning. Our mutual beneficial score describes the significance of contribution from client $i$ to another client $j$ as the difference between the local validation metric score $M_B$ on client $j$ with the model train with client $i$ and the one train without client $i$. Namely,

$$B_{i \to j}^{(t)} := \begin{cases} M_B(\mathbf{w}_{\mathcal{C}\setminus i}^{(t)}, D_V^j) - M_B(\mathbf{w}^{(t)}, D_V^j), & \text{if } M_B \text{ is the lower the better;} \\ M_B(\mathbf{w}^{(t)}, D_V^j) - M_B(\mathbf{w}_{\mathcal{C}\setminus i}^{(t)}, D_V^j) & \text{otherwise,} \end{cases} \tag{4}$$

where $M_B$ can be any of the evaluation metrics, such as validation loss, accuracy, and F1 score. Since there is such value between any pairs of clients, one can represent the influence scores as a matrix $\mathbf{B}^{(t)} \in \mathbb{R}^{N \times N}$, and $\mathbf{B}_{i,j}^{(t)} := B_{i \to j}^{(t)}$. Notice that this matrix is not symmetric in general, because the effect of removing client $i$ on client $j$ is not equivalent to the effect of removing client $j$ on client $i$. Besides, the difference $B_{i \to j}^{(t)}$ can be negative, which means involving client $j$ deteriorates the model performance on client $i$'s validation set. This can happen when the data distributions of these two clients are very different, or there is at least one training dataset is poisoned. Empirically, the naive way to obtain the MBS score is to get $\left\{ \mathbf{w}_{\mathcal{C}\setminus i} \right\}_{i=1}^{N_c}$ and $\mathbf{w}$ directly with conducting $N_c + 1$ times FL runs. If $M_B$ is infinitely differentiable (such as validation loss), the calculation can be further approximated as: $M_B(\mathbf{w}_{\mathcal{C}\setminus i}^{(t)}, D_V^j) - M_B(\mathbf{w}^{(t)}, D_V^j) \approx M_B'(\mathbf{w}^{(t)}, D_V^j)(\mathbf{w}_{\mathcal{C}\setminus i}^{(t)} - \mathbf{w}^{(t)})$, where the two terms can be obtained by one FL trial (Xue et al., 2021). The details defer to Appendix A.

### 3.2 MONETARY QUOTATION

Before the FL starts, each client must submit a monetary quotation. This quotation $b_i \in [0, \infty]$ reflects the profit that client $i$ can obtain with a deployed model that focuses on providing better performance from client $i$'s perspective. Because the monetary quotation has a joint effect with the performance gain, we design to normalize it to a scale comparable to the performance gain:

$v_i = \frac{b_i + \epsilon}{\sum_{j \in C}(b_j + \epsilon)}$. The $\epsilon$ is a small positive constant to validate the extreme case that all clients submit 0 as their quotation. Since there are $N$ clients in a FL task, the equation can be written in a vector form $\mathbf{v} = \frac{\mathbf{b} + \epsilon}{<\mathbf{b} + \epsilon, \mathbf{1}>}$.

## 3.3 ADJUSTABLE WEIGHTS FOR VALIDATION LOSSES

In our framework, both MBS and monetary quotation $v_i$ are encapsulated in the definition of the validation loss aggregation weights, denoted as $\alpha_i$.

$$\alpha_i^{(t)} = v_i \exp\left(\sum_{k=1}^{N} B_{i \to k}^{(t)} / \tau_\alpha\right). \tag{5}$$

The variable $\tau_\alpha$ is the temperature used as a nob to scale the effect of task value and influence score. It is straightforward that $\alpha_i^{(t)}$ positively correlates to the monetary quotation of client $i$ and its contribution to other clients' training. This design ensures that the model favors those clients who make the most performance or monetary contribution and will benefit from the FL. Also, one should expect that $\alpha_i^{(t)}$ can be changed in different iterations because $B_{i \to k}^{(t)}$ usually varies iteration by iteration. The following special cases give some intuition about the effectiveness of the monetary quotation and the performance contribution.

*Equal monetary quotation scenario.* If all the clients submit the same monetary quotation, then the relative magnitudes, $\alpha_i^{(t)} / \alpha_j^{(t)}$, only depend on the contribution of client $i$ and $j$ to other clients. That is, the client that can help others more will be granted with larger weight when selecting hyper-parameters and aggregation weights.

*Data market scenario.* In the scenario of data market, one or multiple data sellers, treated as clients in our FL framework, can submit zero as its monetary quotation. That means that the data sellers give up it most of their privilege in selecting hyper-parameters and aggregation weight. The data buyer can dominate the process and pick the best hyper-parameters and aggregation weights completely for improving model performance on its local dataset.

*Useless user contribution scenario.* If a client's contribution to other parties is very small, i.e., $B_{i \to k}^{(t)} \to 0$, it will only be granted very limited effectiveness on the hyper-parameter and aggregation weight selection without huge monetary contribution. If a client's contribution is negative, then its power on selecting hyper-parameters and aggregation weights will be mostly deprived, because its contribution harms the overall goal.

## 3.4 MARKETFEDHPO

With the definition of the MBS and monetary quotation, the alpha value can be calculated accordingly at the beginning of each communication round. The core idea of our framework is to use $\boldsymbol{\alpha}$ as weights to compose the local validation scores as the training continues. These weighted validation scores guide the model selection process so that the final output model can balance clients' preferences for performance gain and monetary reward. The model selection is achieved by selecting an appropriate combination for two sets of parameters: training hyper-parameters and aggregation weights. The hyper-parameters mainly define how the models should be trained on the client side, including the learning rate and the epoch to local update. Symmetrically, the aggregation weights define how the local models are aggregated. We provide a set of candidates for both hyper-parameter combinations and aggregation weights. Then, FedHPO algorithms can be applied to select the most appropriate hyper-parameter and aggregation weighted for given $\boldsymbol{\alpha}$.

Generally, as shown in Fig. 1, one FL round of MarketFedHPO includes the following procedures. In the beginning, the server distributes the global model to clients. After receiving the global model, the clients' local computation includes local training and local evaluation. Then the updated local models and the necessary metrics for FedHPO are uploaded to the server. When receiving the information from clients, the server first updates the alpha score by Eqn. (5), then updates the hyper-parameters based on the alpha score and the local evaluation results according to the adopted FedHPO methods. Finally, the server aggregates the received local model updates.

Our proposed method is naturally adaptive to a wide range of FL methods and FedHPO methods. As an example, we adapt successive halving algorithm (SHA) and FedEx (Khodak et al., 2021)

as backbones of our model selection algorithm in this paper, because they can explore different hyper-parameter combinations and update the hyper-parameter selection policy on the fly in the training process, and its design embraces the personalization of FL naturally. These two properties echo our needs for the efficiency of the algorithm and deliberate bias toward the client with the highest combined score.

The details of the algorithm are shown in Alg. 1, where $N_m, N_o, N_c$ denotes the number of candidate aggregation weights, candidate local hyper-parameter configurations, and clients, separately. In the algorithm, firstly, a total of $N_m$ models are initialized. Although the initialized models are identical, but each of them will be updated with an associated candidate aggregation weight. At each round, among every model, the FedEx is performed (Line 7-15) to update the configuration distribution $\boldsymbol{\theta}$. Specifically, with the $\boldsymbol{\alpha}$, the hyper-parameter distribution parameter $\theta$ of $m$-th model at $t$-th round is updated by FedEx-Dist-Update, which is defined as follows:

$$\boldsymbol{\theta}_t^{(m)} = \text{Norm}\left(\boldsymbol{\theta}_{t-1}^{(m)} \odot \exp(-\eta_t \tilde{\boldsymbol{\nabla}}_t^{(m)})\right), \tag{6}$$

where Norm denotes the normalization function, and $\text{Norm}(\boldsymbol{\theta}) = \boldsymbol{\theta}/||\boldsymbol{\theta}||_1$; $\odot$ denotes the element-wise multiplication; $\tilde{\boldsymbol{\nabla}}_t^{(m)}$ denotes the gradient approximation of $\boldsymbol{\theta}_t^{(m)}$, whose $j$-th element is calcuated as: $\tilde{\boldsymbol{\nabla}}_t^{(m)}[j] = \frac{\sum_{c=1}^{N_c} \alpha_{t-1}^c (\mathsf{M}_{\text{FedEx}}^{(m,t,c)} - \lambda_t) \mathbf{1}_{o_t^{m,c}=o_j}}{\boldsymbol{\theta}_{t-1}^{(m)}[j] \sum_{c=1}^{N_c} \alpha_{t-1,c}}, \forall o_j \in \mathcal{O}$. When it reaches the SHA elimination round, the models with top $\frac{1}{\eta_{\text{SHA}}}$-quantile performance are selected for the next FedEx and SHA round (Line 20-21). Specifically, the alpha value of each client is calculated at the beginning of each round (Line 4), and is involved in the validation metric calculation of both FedEx and SHA (Line 14 and Line 17).

---

**Algorithm 1** MarketFedHPO

---

**Input:** Aggregation weight candidate set: $\mathcal{O} = \{\mathbf{a}^{(m)}\}_{m=1}^{N_m}$; Local hyper-parameter configuration candidate set: $\{\mathbf{o}_j\}_{j=1}^{N_o}$, schemes for setting step-size $\eta_t$ and baseline $\lambda_t$, Elimination Rate $\eta_{\text{SHA}}$

1: **Server** : Randomly initialize model $\mathbf{w}_0$ and candidate set: $\mathcal{H}_0 = \{\mathbf{w}_0^{(m)} = \mathbf{w}_0\}_{m=1}^{N_m}$;

2: **Server** : Initialize hyper-parameter distribution $\left\{\boldsymbol{\theta}_0^{(m)} = \frac{1}{N_o}\right\}_{m=1}^{N_m}$ ; initialize $\left\{\alpha_{0,c} = \frac{1}{N_c}\right\}_{c \in \mathcal{C}}$;

3: **for** $t = 1, \ldots,$ **do**

4:      **Server** : calculate $\left\{\alpha_{t-1}^{(c)}\right\}_{c \in \mathcal{C}}$ by Eqn. 5

5:   **for** $m \in \left\{m : \mathbf{w}_{t-1}^{(m)} \in \mathcal{H}_{t-1}\right\}$ **do** (In parallel)

6:     **for** $c \in \mathcal{C}$ **do** (In parallel)

7:        **Server** : sample $o_t^{(m,c)} \sim \mathcal{P}\left(\boldsymbol{\theta}_{t-1}^{(m)}\right)$ and share $\left(o_t^{(m,c)}, \mathbf{w}_{t-1}^{(m)}\right)$ to client $c$

8:        **Client** : $\mathsf{M}_{\text{SHA}}^{(m,c,t-1)} \leftarrow \text{Eval}_{\text{SHA}}(\mathbf{w}_{(t-1)}^{(m)}, D_V^c)$;            ▷ Local evaluation

9:        **Client** : $\mathsf{M}_{\text{FedEx}}^{(m,t,c)} \leftarrow \text{Eval}_{\text{FedEx}}(\mathbf{w}_{(t)}^{(m,c)}, D_V^c)$

10:        **Client** : $\mathbf{w}_t^{(m,c)} \leftarrow \text{Train}(\mathbf{w}_{t-1}^{(m)}, D_T^c, o_t^{(m,c)})$,            ▷ Run local training

11:        **Client** : upload $\left(\mathbf{w}_t^{(m,c)}, \mathsf{M}_{\text{SHA}}^{(m,t-1,c)}, \mathsf{M}_{\text{FedEx}}^{(m,t,c)}\right)$

12:     **end for**

13:     **Server** : $\boldsymbol{\theta}_t^{(m)} \leftarrow \text{FedEx-Dist-Update}(\boldsymbol{\theta}_{t-1}^{(m)}, \{\mathsf{M}_{\text{FedEx}}^{(m,t,c)}\}_{c \in \mathcal{C}}, \{\alpha_{t-1}^{(c)}\}_{c \in \mathcal{C}}, \lambda_t, \eta_t)$

14:     **Server** : $\mathbf{w}_t^{(m)} \leftarrow \text{Agg}\left(\left\{\mathbf{w}_t^{(m,c)}\right\}_{c \in \mathcal{C}}, \mathbf{a}^{(m)}\right)$

15:   **end for**

16:   **if** $t \in$ elimination round and $|\mathcal{H}_{t-1}| > 1$ **then**            ▷ Perform SHA

17:     **Server** : $\mathsf{M}_{\text{SHA}}^{(m,t-1)} \leftarrow \sum_{c \in \mathcal{C}} \alpha_{t-1,c} \mathsf{M}_{\text{SHA}}^{(m,t-1,c)}$

18:     **Server** : $\mathcal{H}_t \leftarrow \left\{\mathbf{w}_t^{(m)} : \mathsf{M}_{\text{SHA}}^{m,t-1} < \frac{1}{\eta_{\text{SHA}}}\text{-quantile}\left(\left\{\mathsf{M}_{\text{SHA}}^{(m,t-1)} : \mathbf{w}_{t-1}^{(m)} \in \mathcal{H}_{t-1}\right\}\right)\right\}$

19:   **end if**

20: **end for**

21: Return the remaining $\mathbf{w} \in \mathcal{H}_t$.

---

### 3.5 Payment Schema

The payment happens at the end of the FL training process. Intuitively, the more a client contributes to another client's performance gain on local validation, the more significant position contribution it makes in this FL task. Thus, we design a payment rule so that each client's compensation is proportional to its contribution.

The total amount of monetary rewards that need to be distributed is the sum of all client's quotations for this task. Within this total reward, each client deserves a portion of the reward proportional to its contribution to the other clients' local validation loss. Considering client $i$ has its own monetary quotation $b_i$ for this FL task, the net payment of client $i$ needs to pay is:

$$p_i = b_i - \frac{\exp\left(\sum_{k=1}^N B_{i \to k}^{(t)}/\tau_p\right)}{\sum_{j=1}^N \exp\left(\sum_{k=1}^N B_{j \to k}^{(t)}/\tau_p\right)} \sum_{j=1}^N b_j \quad, \tag{7}$$

where $\tau_p$ is a temperature controlling how fast the portion changes. The payment can be either positive or negative. A positive payment means the client needs to make a monetary contribution to this FL task, while a negative payment means the client can be compensated because of its contribution.

### 3.6 Properties of MarketFedHPO

**Property 1** (Communication and computation cost). MarketFedHPO *requires each client to call the local training or evaluation oracle and share the model at most $\frac{N_m \eta_{SHA} - 1}{\eta_{SHA} - 1} + N_c T$ times.*

If we use SHA (Khodak et al., 2021; Jamieson & Talwalkar, 2016) as the algorithm to select the best aggregation weight, then MarketFedHPO has the following property for providing the best aggregation weights.

**Property 2** (Best aggregation weight). *If 1) $T$ is large enough, 2) MBS and $\mathsf{M}_{SHA}^{m,t}$ converge as $t \to \infty$ and 3) the local hyper-parameter configuration set has size $1$, then Algorithm 1 can ensure the higher price a client pays, the more favorable the final model will be to the client.*

This guarantee directly inherits the guarantee of SHA (Jamieson & Talwalkar, 2016). The key idea is that the more monetary contribution a client makes, the larger influence he can exert on the $\mathsf{M}_{SHA}^{(m,t)}$. More specifically, if a client $c$ increases his monetary contribution so that $\alpha_c > \alpha'_c$ and $\mathsf{M}_{SHA}^{(m,c,t)} > \mathsf{M}_{SHA}^{(m',c,t)}$. If $\mathsf{M}_{SHA}'^{(m,t)} \leq \mathsf{M}_{SHA}'^{(m',t)}$ when client $c$ has $\alpha'_c$, it can happen that $\mathsf{M}_{SHA}^{(m,t)} > \mathsf{M}_{SHA}^{(m',t)}$ if client $c$ increase to $\alpha_c$. But not vice versa. So increasing $\alpha_c$ from $\alpha'_c$ can only increase the probability that less favorable aggregation weight $\mathbf{a}^{(m')}$ filter before a more favorable one $\mathbf{a}^{(m)}$.

Though it is difficult to give a quantitative result of convergence, we will demonstrate in the experiment section that the proposed algorithm can converge very fast and ensure that the favorableness of the final model towards a client is positively correlated with the client's monetary contribution.

## 4 Experiment

In this section, we conduct experiments under various client bid settings to verify the performance control of our proposed method.

**Dataset**. We conduct experiments on CIFAR10 dataset (Krizhevsky, 2009) to verify the performance control of the proposed method. Specifically, to simulate the cross-silo FL scenario, we generate the FL dataset by splitting the dataset to several clients based on the pre-defined class category. More details regarding the FL dataset are in Appendix B.1

**Hyper-parameters**. The search space of the hyper-parameters is listed in Appendix B.2. The metric to calculate the influence score is the validation loss, the metric adopted in FedEX is validation loss, and the metric in SHA of aggregation weight is validation F1 score.

**Client bids**. To simulate the scenario where different willingness to performance gain and monetary of clients in FL, i.e., some clients want to obtain more performance gain from the model, while some clients want to receive more payment, we set different bids for clients. For example, when the client bids are 10:0:0, it means that client 1 wants to achieve a higher performance gain, while the other clients prefer montary rewards.

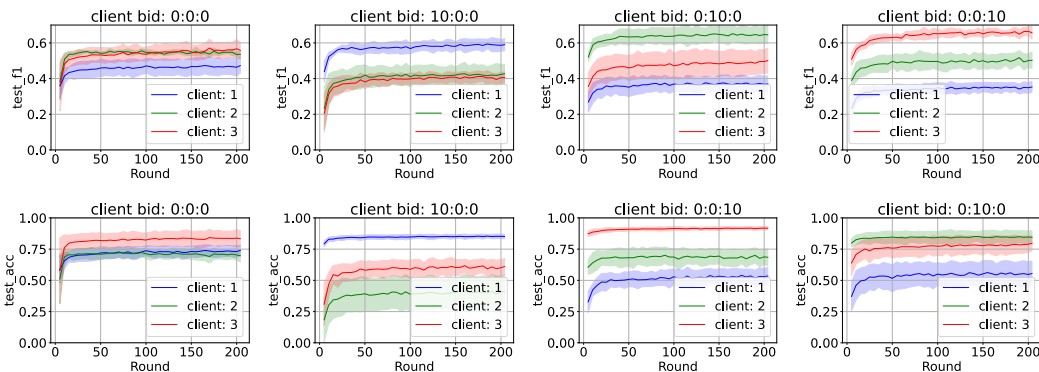

(a) Setting: Three clients, with one client prefers performance gain and others prefer monetary rewards.

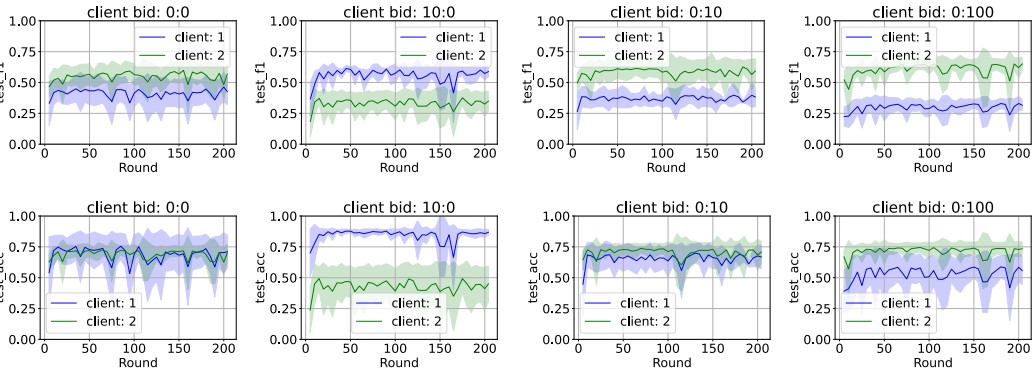

(b) Setting: Two clients,with one client prefers performance gain and others prefer monetary rewards.

Figure 2: Performance under different client bids.

## 4.1 RESULTS

We report the accuracy and the F1 score of each clients. Fig. 2 shows the performance of each client. We report the mean and standard deviation with 5 times repeated runs (with different random seeds). It can be observed that compared with the case where clients have the same bids, the client can gain a better performance when the corresponding bid increases. Specifically, comparing column 1 and column 2 in Fig. 2(a), we observe a significant improvement in the model's performance on client 1's data when their bid is increased to 10, while the bids of the other clients remain at 0. Furthermore, when client 1 increases their bid to become the highest among the three clients, the model's performance on their data changes from being the worst to the best compared to the other clients. The same trend can be found in the other two clients. Similarly, in the two-clients dataset, the same trend can be found in Fig. 2(b). These observations validate the efficacy of our proposed method, as the rise in bids signifies the client's willingness to acquire a model with superior performance at the expense of increased monetary investment. More experimental results with clients bids selected from set $\{10, 20, 50\}$ are in Appendix C.2

The experiments demonstrate that our method can effectively align the client's performance with their bids, which reflect their preference and budget. By increasing their bids, the clients can obtain a model that performs better on their own data, while sacrificing some performance on the other clients' data. This shows that our method can achieve a fair and flexible trade-off between the clients' willingness and the model performance.

**Study about Alpha Value.** Fig. 3 shows the calculated alpha value. The trend of the alpha value is consistent with the performance shown in Fig. 2. This indicates that our proposed method empowers clients to invest more funds in achieving a higher alpha value, ultimately leading to the final model being more beneficial to their unique data.

**Study about Payment**. Fig. 4 shows the averaged payments and performance results of clients under

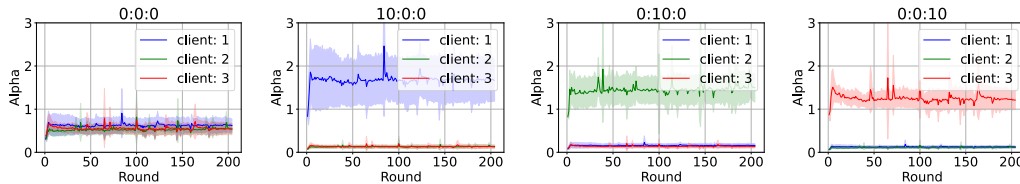

(a) Setting: Three clients. Each title of the subfigure denotes the client bids.

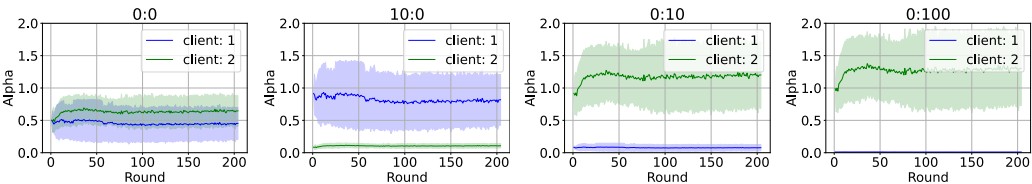

(b) Two clients. Each title of the subfigure denotes the client bids.

Figure 3: Alpha value under different client bids.

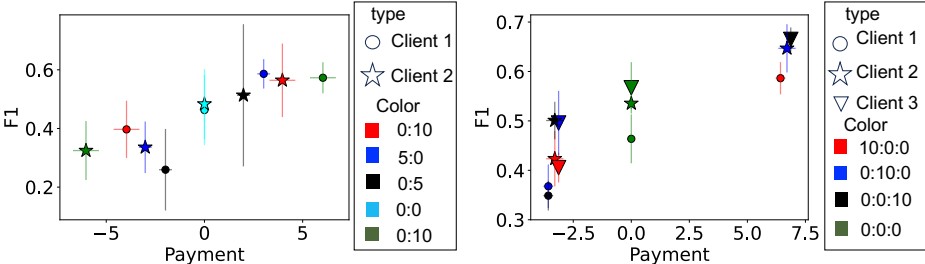

Figure 4: Payment and F1. Left: Two Clients; Right: Three Clients. The error bar represents the standard deviation.

different client bids. The positive value of payment indicates that the client gives the payment to others, and the negative value indicates that the client receives payment from others. The dots with the same colour represent one group of clients in one bid, and the markers with the same marker represent one client. For the purpose of clarity, in the figure, we show the payment and the F1 score with bids around 10. From the figure, it can be observed that within one-time bid, the client with the highest price pays the highest value to others and obtains the best performance. The other clients whose bids are 0 indicating their high request of monetary reward, receive the payment based on their corresponding data contribution. This observation demonstrates the ability of our proposed method in adjusting different reward type preferences.

## 5 CONCLUSIONS

Federated Learning (FL) has emerged as a promising solution for collaborative model training from isolated data sources without the direct sharing of private data. The incentive mechanism plays a crucial role in motivating clients to participate and contribute data. Most existing incentive mechanisms take a post-hoc form, which may not be flexible in cases where clients act as both data contributors and model buyers with different preferences. To address this challenge, our proposed framework Alpha-Tuning, embeds an adaptive incentive mechanism that prioritizes model performance of clients who have made higher overall contributions, based on their monetary and data contributions. By using an overall contribution-based hyper-parameter selection criteria and a data contribution-based payment schema, this approach offers a flexible and efficient solution to satisfy clients' various preferences towards model performance gains and monetary rewards. Overall, this paper contributes to the advancement of FL frameworks with an incentive mechanism that can adaptively adjust different clients' preferences.

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
