# A    MUTUAL BENEFICIAL SCORE

*Comparing LOO with Shapley value.* Another popular metric for data valuation is Shapley value (SV) (Ghorbani & Zou, 2019; Wang et al., 2020; Liu et al., 2022). However, the SV may introduce large computation overhead; if some approximation techniques are used, the communication cost need to trade-off with the estimation accuracy (Wang et al., 2020). Moreover, SV reflects the influences of any subset of clients on global performance. However, how one client's contribution affects the model performance on another client's local dataset is unclear with SV. Following the prvious example, if Client A and B have the same data by chance, they should have about the same SV. However, it is hard to tell the reason for the same SV, whether it is because Client A and B share the same data or their data are different but contribute equally to the model. For the former case, Client B should not be rewarded by Client A. On the contrary, our proposed MBS can provide explicit beneficial relations between clients to guide the monetary flow while guaranteeing efficiency with a straightforward core idea without the need for tuning trade-offs.

The details of the directly get the $\left\{\mathbf{w}_{\mathcal{C} \setminus i}\right\}_{i=1}^{N_c}$ and $\mathbf{w}$ are shown in Alg. 2.

---

**Algorithm 2** Mutual beneficial score (MBS)

---

**Input:** Hyper-parameter distribution set $\Theta = \{\boldsymbol{\theta}_m\}_{m=1}^{N_m}$
**Output:**

1: initialize the MBS model set: $\mathcal{H} = \left\{\mathbf{w}_m^{(0)}\right\}_{m=1}^{N_m}$
2: **for** $t = 1, \ldots, T$ **do**
3:     Server: send $\mathbf{w}_m^{(t-1)}$ to clients
4:     **for** $c = 1, \ldots, N_c$ **do**                                   ▷ In parallel
5:         **for** $m = 1, \ldots, N_c + 1$ **do**
6:             Client: update $\mathbf{w}_m^{(t-1)}$ to $\mathbf{w}_{m,c}^{(t)}$ by local training
7:             Client: send updated model $\mathbf{w}_{m,c}^{(t)}$, updated model's evaluation $M_{\text{FedEx}}(\mathbf{w}_{t,c}^{(m)})$, and
      global model's evaluation $M_B(\mathbf{w}_m^{(t-1)}, D_V^c)$ to Server
8:         **end for**
9:     **end for**
10:     Server: update $\Theta$ by FedEx.                    ▷ Update local training FedEx
11:     Server: $\mathbf{w}_m^{(t)} \leftarrow \text{Agg}\left(\left\{\mathbf{w}_{m,c}^{(t)}\right\}_{c \in \mathcal{C}, c \neq m}\right)$
12:     Server: Update $\left\{B_{t-1}^{i,j}\right\}_{i=1,j=1}^{N_c,N_c}$ with Eqn. 4
13: **end for**

---

# B    EXPERIMENT SETTINGS

## B.1    DATASET

The pre-defined class category contains the information of the clients regarding the classes it owns most. With pre-defined class category, the data splitter is as follows. Suppose in the pre-defined class category, client $\#c$ owns class set $\mathcal{K}$. If sample $s$'s label belongs to $\mathcal{K}$, it has the probability of $1 - p$ to in client $c$ and probability of $\frac{p}{|\mathcal{C}|-1}$ in client within $\mathcal{C} \setminus c$, where $\alpha$ is the heterogeneous ratio. We set the client number $|\mathcal{C}|$ to 2 and 3, separately. The $p$ value and the class categories of each clients are shown in Table 1. The statistics of the FL datasets with two and three clients, separately, are listed in Table 2

| | Dataset with two clients ($p = 0.01$) | | Dataset with three clients ($p = 0.05$) | | |
|---|---|---|---|---|---|
| Client ID | Client #1 | Client #2 | Client #1 | Client #2 | Client #3 |
| Class set | 0, 1, 8, 9 | 2, 3, 4, 5, 6, 7 | 0, 1, 2, 3 | 4, 5, 6 | 7, 8, 9 |

Table 1: Class category across clients.

| | Dataset with two clients ($\alpha = 0.01$) | | | | | | Dataset with three clients ($\alpha = 0.05$) | | | | | | | | |
| | Client #1 | | | Client #2 | | | Client #1 | | | Client #2 | | | Client #3 | | |
| Class | train | valid | test | train | valid | test | train | valid | test | train | valid | test | train | valid | test |
|---|---|---|---|---|---|---|---|---|---|---|---|---|---|---|---|
| 0 | 3985 | 964 | 990 | 41 | 10 | 10 | 3824 | 926 | 950 | 100 | 24 | 25 | 102 | 24 | 25 |
| 1 | 3960 | 990 | 990 | 40 | 10 | 10 | 3800 | 950 | 950 | 100 | 25 | 25 | 100 | 25 | 25 |
| 2 | 39 | 10 | 9 | 3889 | 1062 | 991 | 3731 | 1017 | 950 | 98 | 27 | 24 | 99 | 28 | 26 |
| 3 | 40 | 10 | 10 | 3980 | 970 | 990 | 3819 | 931 | 950 | 100 | 24 | 25 | 101 | 25 | 25 |
| 4 | 39 | 10 | 9 | 3955 | 996 | 991 | 99 | 24 | 25 | 3794 | 956 | 950 | 101 | 26 | 25 |
| 5 | 40 | 9 | 9 | 3972 | 979 | 991 | 100 | 25 | 25 | 3811 | 938 | 950 | 101 | 25 | 25 |
| 6 | 40 | 9 | 9 | 3966 | 985 | 991 | 100 | 25 | 25 | 3805 | 943 | 950 | 101 | 26 | 25 |
| 7 | 40 | 10 | 10 | 3960 | 990 | 990 | 100 | 25 | 25 | 100 | 25 | 25 | 3800 | 950 | 950 |
| 8 | 3960 | 990 | 990 | 40 | 10 | 10 | 100 | 25 | 25 | 100 | 25 | 25 | 3800 | 950 | 950 |
| 9 | 3973 | 975 | 990 | 41 | 11 | 10 | 100 | 24 | 25 | 100 | 25 | 25 | 3814 | 937 | 950 |

Table 2: The statistics of two FL datasets.

## B.2 HYPER-PARAMETERS

The search space of the hyper-parameters is listed in Table 3. The metric to calculate the influence score is the validation loss, the metric adopted in FedEX is validation loss, and the metric in SHA of aggregation weight is validation F1 score.

| Name | Search space |
|---|---|
| Learning rate | [0.01, 0.02, 0.04, 0.06, 0.08, 0.1, 0.2, 0.4, 0.6, 0.8] |
| Local update steps | [1, 2, 3, 4] |
| Aggregation weights (two clients) | [0.9, 0.1], [0.8, 0.2], [0.3, 0.7], [0.6,0.4], [0.4, 0.6],[0.5, 0.5], [0.2, 0.8], [0.7, 0.3], [0.1, 0.9], [0.0, 1.0], [1.0, 0.0] |
| Aggregation weights (three clients) | [0.7, 0.1, 0.2], [0.3,0.3,0.4], [0.1, 0.2, 0.7], [0.1,0.7, 0.2], [0.5, 0.3, 0.2], [0.2,0.5,0.3], [0.2, 0.3,0.5] |

Table 3: Hyper-parameter search space.

## C EXPERIMENT RESULTS

### C.1 AGGREGATION WEIGHTS

Table C.1 lists the selected aggregation weights under 5 times repeated runs with different random seeds. It can be observed that the higher the client bid is, the corresponding aggregation weights prone to have a higher value.

| | Client bids | Aggregation weights (5 times with different random seeds) |
|---|---|---|
| Three clients | [0, 0, 0] | [0.5, 0.3, 0.2], [0.3, 0.3, 0.4], [0.3, 0.3, 0.4], [0.3, 0.3, 0.4], [0.3, 0.3, 0.4]. |
| | [10, 0, 0] | [0.5, 0.3, 0.2], [0.7, 0.1, 0.2], [0.7, 0.1, 0.2], [0.7, 0.1, 0.2], [0.8, 0.1, 0.1]. |
| | [0, 10, 0] | [0.1, 0.7, 0.2], [0.3, 0.3, 0.4], [0.2, 0.5, 0.3], [0.2, 0.5, 0.3], [0.1, 0.8, 0.1]. |
| | [0, 0, 10] | [0.1, 0.2, 0.7], [0.1, 0.2, 0.7], [0.2, 0.3, 0.5], [0.1, 0.1, 0.8], [0.2, 0.3, 0.5]. |
| Two clients | [0, 0] | [0.3, 0.7], [0.2, 0.8], [0.4, 0.6], [0.6, 0.4], [0.6, 0.4]. |
| | [0, 10] | [0.6, 0.4], [0.2, 0.8], [0.4, 0.6], [0.2, 0.8], [0.3, 0.7]. |
| | [10, 0] | [0.7, 0.3], [0.7, 0.3], [0.7, 0.3], [0.8, 0.2], [0.7, 0.3]. |
| | [0, 100] | [0.3, 0.7], [0.1, 0.9], [0.4, 0.6], [0.3, 0.7], [0.2, 0.8], |

Table 4: The selected aggregation weight by SHA.

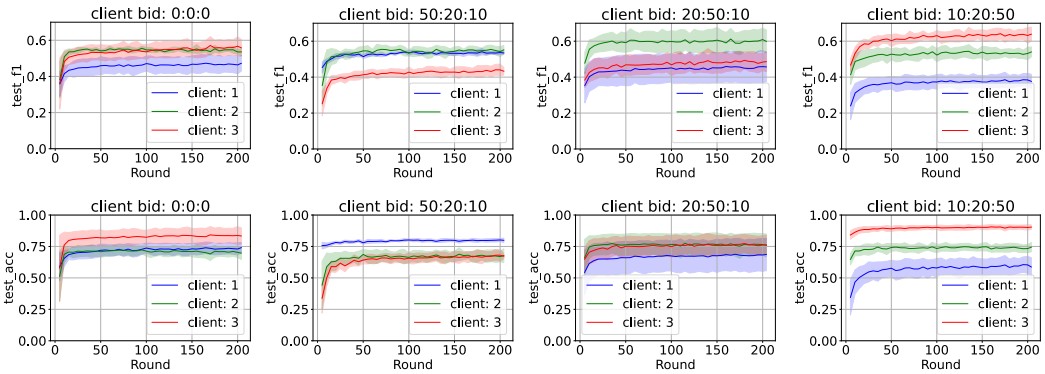

Figure 5: Three clients with their bids selected from $\{10, 20, 50\}$

## C.2 DIFFERENT CLIENT BIDS.

In the previous study of three client settings, only one client has the high preference of performance gain. Here we study the case that three clients all report the bid greater than zero, indicating different preference level to the performance gain. Fig. 5 shows the performance under different clients bids, and the trend similar to Fig. 2(a) can be observed. The final performance and the clients' payment are listed in Table 5.

In the bid of $0 : 0 : 0$, it can be observed that client 1 has the lowest performance without additional monetary investment. This also indicates the highest difficult of improving client's performance, because as shown in Fig. 3(a), the three clients have the similar alpha, indicating the similar data contribution. It is reasonable that client 1 only have the comparable performance with the second best client under bids $50 : 20 : 10$, comparing with the bids of $20 : 50 : 10$ and $10 : 20 : 50$ that the client who offers the highest bid gains a much superior performance than the second best client. Therefore, the client with the most difficult data should pay more to get the best results, referring the result of bid $100 : 20 : 10$ in Table 5.

| Client bids | Client 1 | | Client 2 | | Client 3 | |
|---|---|---|---|---|---|---|
| | Payment | F1 | Payment | F1 | Payment | F1 |
| 100:20:10 | $53.537 \pm 2.775$ | $0.549 \pm 0.022$ | $-22.834 \pm 4.854$ | $0.520 \pm 0.078$ | $-30.703 \pm 2.291$ | $0.426 \pm 0.038$ |
| 20:50:10 | $-8.593 \pm 1.708$ | $0.458 \pm 0.090$ | $23.640 \pm 2.987$ | $0.609 \pm 0.063$ | $-15.048 \pm 1.410$ | $0.484 \pm 0.048$ |
| 50:20:10 | $21.407 \pm 1.708$ | $0.537 \pm 0.011$ | $-6.360 \pm 2.987$ | $0.542 \pm 0.028$ | $-15.048 \pm 1.410$ | $0.440 \pm 0.032$ |
| 10:20:50 | $-18.593 \pm 1.708$ | $0.384 \pm 0.036$ | $-6.360 \pm 2.987$ | $0.532 \pm 0.035$ | $24.952 \pm 1.410$ | $0.632 \pm 0.044$ |
| 0:0:0 | $0.000 \pm 0.000$ | $0.464 \pm 0.050$ | $0.000 \pm 0.000$ | $0.535 \pm 0.011$ | $0.000 \pm 0.000$ | $0.567 \pm 0.052$ |

Table 5: Averaged Payment and F1 score under different client bids.

## C.3 ABLATION STUDY

### C.3.1 FEEDBACK METRIC

Here we study the robustness of the proposed method under different feedback metric settings. In our proposed method, there are three parts containing the feedback metric selection: the MBS calculation, local training FedEx and aggregation weights SHA. In the previous section, we present the results with the feedback metrics adopted are: average validation loss for both MBS and FedEx, and F1 score for aggregation weight SHA. In this part, we examine the performance changes under 2 different settings: (1) average validation loss for MBS, FedEx, and aggregation weight SHA; (2) average validation loss for FedEx, and F1 socre for MBS and aggregation weight SHA.

Fig. 6 and Fig. 7 show the results under the above two different feedback metric settings, separately. Similar trend that the higher the bid, the better the performance is observed. Besides, the rank of the

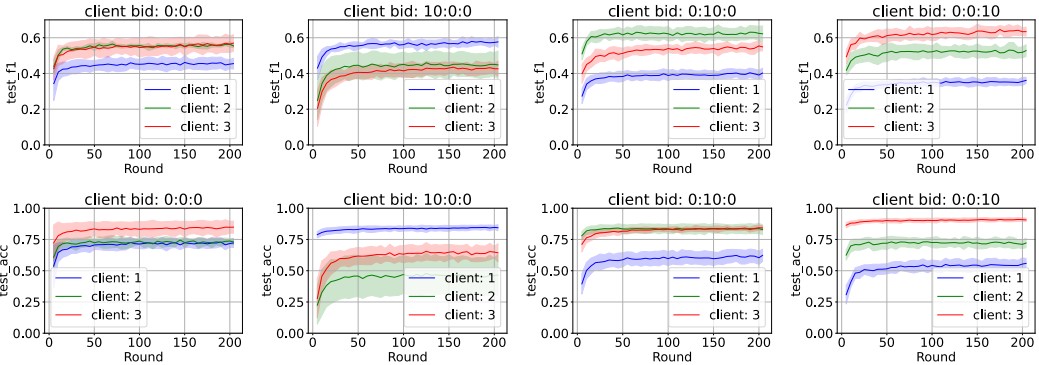

(a) Setting: Three clients,with one client prefers performance gain and others prefer monetary rewards.

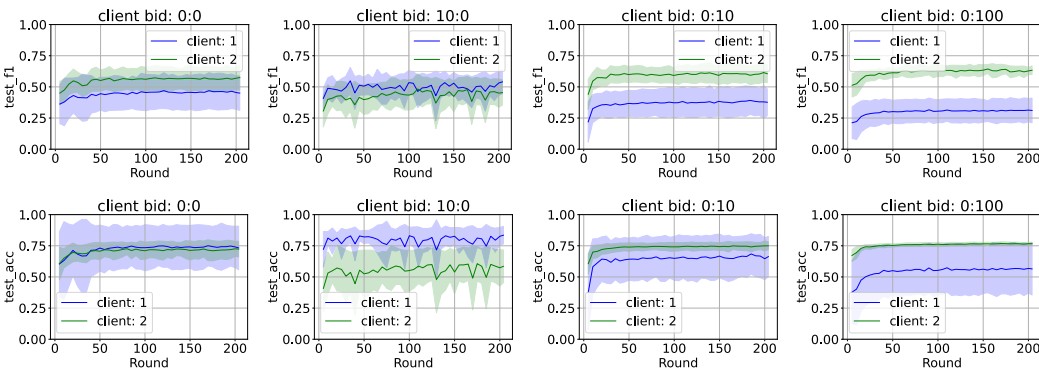

(b) Setting: Two clients,with one client prefers performance gain and others prefer monetary rewards.

Figure 6: Results with under feedback metrics: average validation loss for MBS, FedEx, and aggregation weight SHA.

client in the bid is also preserved in their performance rank. Those observations demonstrate the robustness of our proposed method under different feedback metrics.

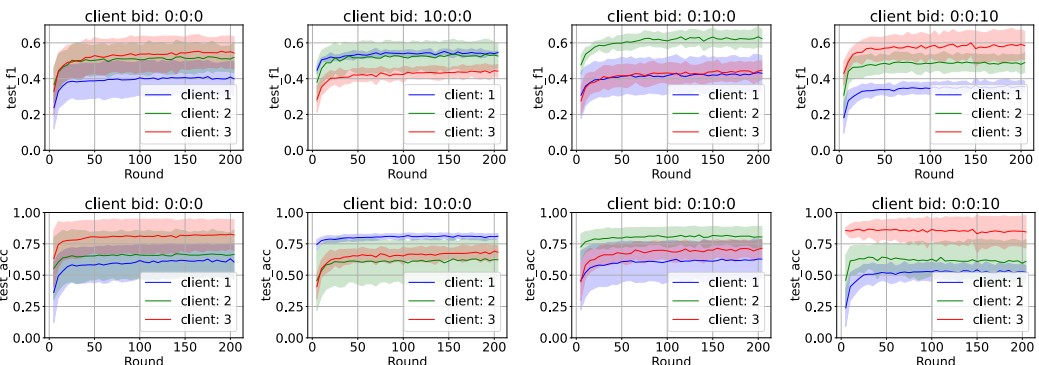

(a) Setting: Three clients,with one client prefers performance gain and others prefer monetary rewards.

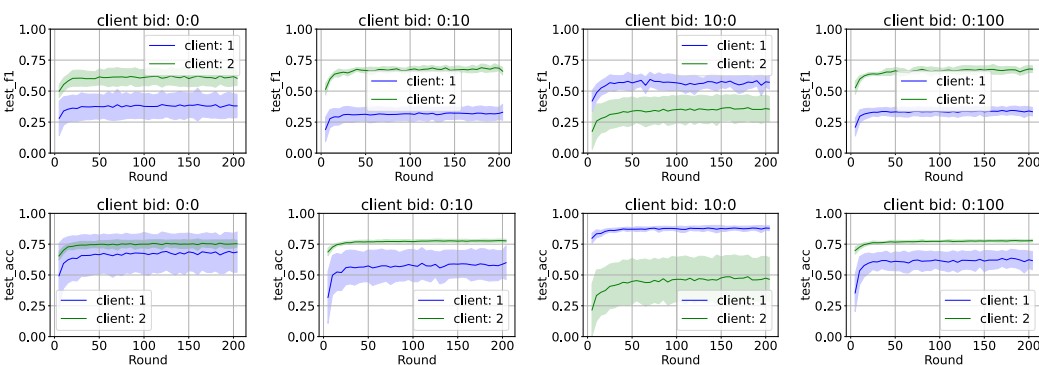

(b) Setting: Two clients,with one client prefers performance gain and others prefer monetary rewards.

Figure 7: Results with under feedback metrics: average validation loss for MBS, and F1 score for FedEx and aggregation weight SHA.

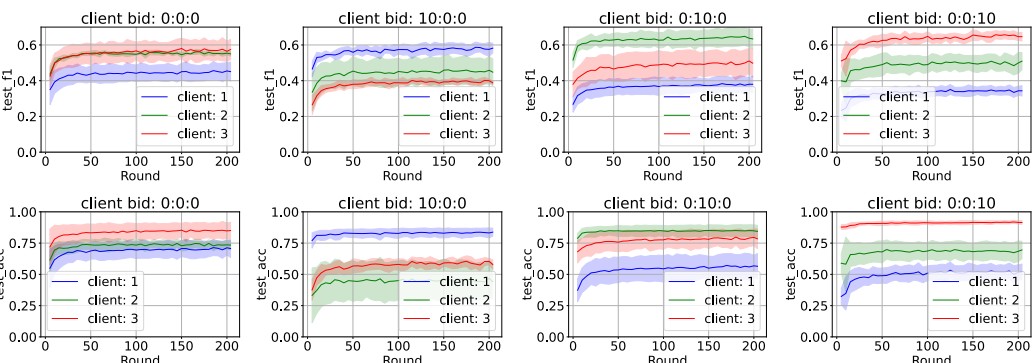

(a) Setting: Three clients,with one client prefers performance gain and others prefer monetary rewards.

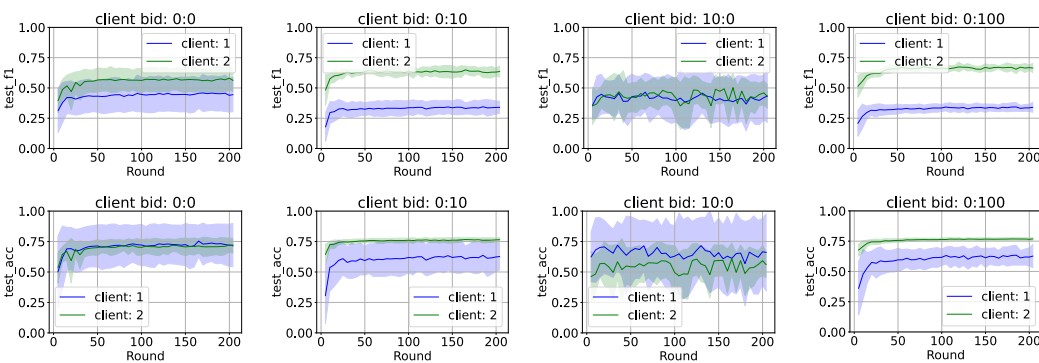

(b) Setting: Two clients,with one client prefers performance gain and others prefer monetary rewards.

Figure 8: Results with feedback metrics: average validation loss for MBS, FedEx, and F1 for aggregation weight SHA.