# OpenReview forum: "Performance Adjustment for Federated Learning Marketplace"
_ICLR.cc/2024/Conference — Submitted to ICLR 2024_

### Official Review · Reviewer_ijEK · 2023-10-18

**Soundness:** 2 fair
**Presentation:** 2 fair
**Contribution:** 2 fair
**Rating:** 3
**Confidence:** 4

**Summary:**

This paper proposes a FL framework embedded with an incentive mechanism. It balances the preferences of clients (double roles: model trainer and model user) to model performance gain and monetary reward by designing the aggregation weight $\alpha$ for local models. The aggregation weight $\alpha$ consists of MBS and monetary quotations. The MBS defines one client’s contributions on model performance to others. And the monetary quotations indicate that one client’s will to pay for model performance gains. Clients with low-quality data may provide a higher monetary quotation to make its aggregation weight $\alpha_i$ be larger so to get more model gains. Clients with more data (data seller) do not care about model performances but hope to receive monetary rewards through contributes better model performances based on more local data.

**Strengths:**

1.	This paper is well organized and easy to follow.
2.	This paper considers the double roles of FL clients and propose an incentive mechanism to balance the preferences of clients to model performance gain and monetary reward, which is an important and interesting topic in federated learning.
3.	The proposed algorithm is comprehensive and practical, since it designs the model aggregation weights based on model performance and monetary quotations and consider the adjustments of hyper-parameters through typical FedHPO methods.

**Weaknesses:**

1.	The proposed two metrics (MBS and monetary quotations) are commonly used and not novel, especially the latter is commonly used in prior auction-based studies, as the authors stated in related work (page 2).
2.	There are some notations confusing. Such as:
-	Page 4: “which means involving client j deteriorates the model performance on client i’s validation set”, i and j may be exchanged.
-	Page4: what does $N_C$ mean? Does it equal to $N$?
3.	In page 4, the simplified MBS calculation uses the derivative of evaluation metric, what is computed and denoted in experiments? I can not find the illustration for this clarification.
4.	In page 4, Section 3.2 Monetary quotation, the second sentence confused me. The quotation $b_i$ should be the highest willing payment of client $i$ to the FL system, not the profits received from others. The authors are suggested to re-organize the description.
5.	The theoretical analysis for convergence is necessary for guaranteeing the effectiveness of FL algorithm.
6.	Experiments are not enough to verify the advance of the proposed method due to the following reasons:
-	Only one small CIFAR-10 dataset is used for experiments. The more and larger dataset should also be used for testing.
-	There are only two or three clients allocated in the FL system. As we all know, a cross-device FL system often many participants, so the effectiveness of the proposed method in a FL system with more clients should be evaluated.
-	No prior baselines are compared, so the advance of the proposed method is unknow.
-	Figure 2 (a), the last two figures in the bottom line are reversed.
-	Figure 2 (b), client bid: 0:10, why tesy_acc improves far less than test_f1?

**Questions:**

Please see above

---

### Official Review · Reviewer_Eroa · 2023-10-27

**Soundness:** 2 fair
**Presentation:** 3 good
**Contribution:** 2 fair
**Rating:** 3
**Confidence:** 4

**Summary:**

This paper introduces a federated learning framework, Alpha-Tuning, that motivates from client incentives and adaptively adjusts client model performances based on both data and monetary contributions. The paper formulates the client incentivization problem into a federated hyper-parameter optimization (FedHPO) problem where the optimal aggregation weights are updated based on the client contributions. As such, previous works on FedHPO are directly adopted here. The method also includes the distribution of monetary rewards back to all clients. The authors are able to demonstrate the efficacy of the proposed mechanism/framework through experiments with 3 federated clients possessing CIFAR-10 data.

**Strengths:**

1. The incentive-related motivation in FL is important and interesting.
2. The recognition of the two types of federated clients with different objectives in monetary compensation and model performance improvements is valid.

**Weaknesses:**

1. The denomination between monetary quotation and model improvement is not designed and analyzed carefully.
2. The incentives for clients to join the federated effort were not carefully discussed.
3. The design choices for SHA and FedEx need to be better justified.
4. The empirical experiments and evaluations are not comprehensive.

More details about the weaknesses are elaborated below in the Questions.

**Questions:**

1. The denomination between monetary quotation and model improvement is not designed and analyzed carefully. As a high-level question, how much money should correspond to the model performance increment from 90% to 91% for a specific client’s validation data?
2. From Eq. (7), I can see that monetary quotations from all clients are distributed back to clients depending on the contributions $B$. However, this monetary reward is not linked to performance guarantees and may make clients lose incentive. To elaborate, even if the monetary quotation is high for a client (i.e., the client is willing to spend much money for an improved performance), but it is entirely possible that the model for the client cannot be further improved (e.g., one possibility is that other clients’ data is not helpful). In this case, the money quotation from the specific client still gets distributed to others, when there is no model improvement. How do you justify this case under your design?
3. Firstly, I believe there is a typo for the green legend in Figure 4 (left): The green label should be for “10:0” right? From this figure, we observe that client 1 gets a lower F1 score when the quotation is 10 (green circle) as compared to when the quotation is 5 (blue circle). I.e., client 1 gives more money but gets a worse model. This reinforces my Question 2 about having no guarantee for model performance with respect to monetary quotation. Could you justify this case? Could you explain the client’s incentives and behaviors, too?
4. In the paper, the training aggregation weights $\beta_i$ are treated as hyper-parameters. However, they are closely linked to the final model trained as well. Why can’t they be considered as designs to favor specific clients, just like what $\alpha$ does?
5. There is no convergence result for the algorithm proposed. In fact, the authors wrote in section 3.3 that “$\alpha_i^{(t)}$ can be changed in different iterations”. How can you assure convergence when the loss (or, objective) is constantly changing? Does the objective converge as well? If no theoretical convergence can be given, then the paper becomes a lot less convincing.
6. The design choices for SHA and FedEx need to be better justified. Could you elaborate on why these methods fit your specific incentive mechanism? Or, any other FedHPO method could possibly work too?
7. Wouldn’t a client benefit a lot more by giving a very difficult, or potentially adversarial (e.g., wrong label) validation points? In this case, the other clients are unlikely to improve on this client’s validation dataset. Therefore, this client would not need to pay other clients (or pay very little) and obtain more money back. Even worse, when every client is now incentivized to state adversarial validation datasets, the federated training process can be adversely affected.
8. The assumptions 2) and 3) in Property 2 sound very stringent to me, it might make the property much less useful in practice.
9. The empirical experiments are not comprehensive enough. The results are not convincing with only 3 clients, could you produce results for at least 50 federated clients with data splits following standard heterogenous FL baselines/benchmarks? The dataset used (CIFAR-10) is also very well-behaved. Can you show experiments with larger-scale datasets like ImageNet and Sentiment-140?
10. I see in Table 3 of Appendix B.2 that the aggregation weights candidates are a discrete set of potential values. Could you optimize a continuous vector for it instead?
11. There are existing works that have looked into model incentives (instead of monetary incentives), but none of them are cited or discussed in the paper. One example would be [1].

[Minor]

12. It should be made clear that (3) is not a direct reduction of (2). Please correct me if I am wrong.
13. In the fourth line of the “Computation of MBS.” paragraph, there is a typo. It should be $w_{\mathcal{C} \setminus i}^{(t)}$ instead of $w_{\mathcal{C} / i}^{(t)}$.
14. What is the reason for creating an influence score matrix $\mathbf{B}$ after Eq. (4)? You are going to add them over the clients anyway, i.e., in Eq. (5).

**References**

[1] Gradient-Driven Rewards to Guarantee Fairness in Collaborative Machine Learning. NeurIPS, 2021.

---

### Official Review · Reviewer_C98P · 2023-11-01

**Soundness:** 2 fair
**Presentation:** 2 fair
**Contribution:** 2 fair
**Rating:** 3
**Confidence:** 4

**Summary:**

This work analyzes clients' incentives in federated learning where clients have cost to participate in FL and get corresponding monetary benefits from the final global model. The work proposes to give different weights to clients depending on their local validation loss which can translate to the client's performance contribution in the given training round. Gaining intuition from the leave-on-out method, the work proposes to train $N+1$ model where one model is trained with all clients and all the other models are trained with updates from the $N-1$ clients' update. They show that such method can satisfy different clients' participation FL with different quotation settings and costs.

**Strengths:**

- The work investigates a relevant problem in FL where clients' incentives and their corresponding participation cost and compensation are considered.

- The work incorporates an interesting approach of hyper-parameter tuning + leave-one-out method to modulate the performance monetary quotation and compensation for the clients based on their validation losses.

- The work provides detailed explanation of their method for the readers to understand including ideas in which the work was originated from with relevant references.

**Weaknesses:**

- The main issue I have with the work is its feasibility to be implemented in realistic FL settings. For instance, how likely is that the monetary quotation to be fixed throughout the training rounds for the clients in the FL setting and also how likely is it that it is feasible to train $N+1$ models in FL? With increasingly high computation and communication cost of training models via FL increasing the number of modes seem to be infeasible and rather unrealistic.

- Another weakness I can see from the FedHPO method is that the payment happens at the end of the FL training, depending on how a client contributes to another clients' performance. However, such performance is not deterministic, and may change throughout the course of training. In this case, how would the clients know in advance of whether their quotations are met or the monetary contribution is positive? If the clients don't have such guarantees, it will be easier for them to simply opt-out of the training process.

- Lastly, the empirical validation of the work seems weak. Since the work is mainly empirical, I think a more thorough evaluation is needed including more clients and different tasks. In the current version, only limited number of clients are considered with the CIFAR10 dataset.

**Questions:**

See weaknesses above.

---

### Official Review · Reviewer_1CDp · 2023-11-07

**Soundness:** 2 fair
**Presentation:** 2 fair
**Contribution:** 2 fair
**Rating:** 5
**Confidence:** 4

**Summary:**

This work introduces an Alpha-Tuning framework, which addresses the challenge of optimizing federated learning (FL) processes to align model performance and monetary rewards with varying client preferences.
At the core of Alpha-Tuning is a mechanism for dynamically determining the weights assigned to clients' local validation losses.
These weights are based on each client's performance contribution in the training round and its monetary quotation, allowing for a personalized adjustment of FL course bias.
In addition, this submission includes a payment rule designed to compensate clients according to their data contribution.
This ensures that clients are rewarded in a way that aligns with their level of participation in the FL process.
The effectiveness of the Alpha-Tuning framework is demonstrated through experiments on CIFAR10.

**Strengths:**

The motivation is clearly articulated and compelling.

The discussion and illustrations are generally accessible, but there is room for improvement in enhancing clarity and comprehensibility.

**Weaknesses:**

The primary concern with this submission pertains to the experimental comparison:
- There appears to be a lack of comparison between the proposed solution and prior works.
- The experiments rely solely on a single dataset, CIFAR-10, making it challenging to assess the effectiveness of the proposed solution comprehensively.
- Additionally, the parameter settings in the experiments raise some questions. For instance, the number of clients seems relatively small. In prior works, using 20 or 100 nodes is a more common and reasonable choice for evaluating federated learning methods.

**Questions:**

Please refer to "Weaknesses".

---

### Meta-Review · Area_Chair_Nfn8 · 2023-12-06

**Metareview:**

This paper introduces a Federated Learning (FL) framework that incorporates an incentive mechanism to balance the preferences of clients who play dual roles as both model trainers and users. The proposed approach achieves this by designing the aggregation weight for local models. This weight includes both Model-Based Scores (MBS) and monetary quotations, where MBS represents one client's contributions to the model's performance for others, and monetary quotations signify a client's willingness to pay for improvements in model performance. However, the distinction between monetary quotations and model improvement lacks careful design and analysis.

The paper falls short in discussing the incentives for clients to actively participate in the federated effort, and the empirical experiments and evaluations are insufficient. The absence of a response from the authors further compounds these concerns. Consequently, I recommend rejecting this paper.

**Justification For Why Not Higher Score:**

N/A

**Justification For Why Not Lower Score:**

N/A

---

### Decision · Program_Chairs · 2024-01-16

Reject